# Upfront Surgery vs. Endoscopic Stenting Bridge to Minimally Invasive Surgery for Treatment of Obstructive Left Colon Cancer: Analysis of Surgical and Oncological Outcomes

**DOI:** 10.3390/cancers16233895

**Published:** 2024-11-21

**Authors:** Mauro Marzano, Paolo Prosperi, Gian Luca Grazi, Fabio Cianchi, Luca Talamucci, Damiano Bisogni, Lapo Bencini, Manuela Mastronardi, Tommaso Guagni, Agostino Falcone, Jacopo Martellucci, Carlo Bergamini, Alessio Giordano

**Affiliations:** 1Emergency Surgery Unit, Careggi University Hospital, Largo Brambilla 3, 50134 Florence, Italyprosperip@aou-careggi.toscana.it (P.P.); martelluccij@aou-careggi.toscana.it (J.M.);; 2Hepatobiliary Pancreatic Unit, Careggi University Hospital, Largo Brambilla 3, 50134 Florence, Italy; gianluca.grazi@unifi.it; 3Digestive System Surgery Unit, Careggi University Hospital, Largo Brambilla 3, 50134 Florence, Italy; fabio.cianchi@unifi.it; 4Advanced Interventional Endoscopy Unit, Careggi University Hospital, Largo Brambilla 3, 50134 Florence, Italy; talamuccil@aou-careggi.toscana.it (L.T.); bisognifelice@libero.it (D.B.); 5General Surgery Unit, Careggi University Hospital, Largo Brambilla 3, 50134 Florence, Italy; bencinil@aou-careggi.toscana.it; 6Department of Medicine, Surgery and Health Sciences, General Surgery Unit, University Hospital of Trieste, Strada di Fiume 447, 34149 Trieste, Italy

**Keywords:** colorectal cancer, minimally invasive surgery, endoscopic stenting, left colon cancer, intestinal obstruction

## Abstract

Obstructive left colon cancer is considered a surgical emergency. Therapeutic options include tumour resection (with or without anastomosis) or stent placement to alleviate the obstruction and prepare the patient for elective resection surgery (staged resection). However, there is no consensus on the appropriate treatment. Stents have shown a good safety profile regarding surgical outcomes, but concerns remain on their oncological effects. This study analyses surgical and oncological outcomes in a cohort of patients treated for obstructive left colon cancer through primary resection or stent placement followed by staged resection. Patients undergoing staged resection showed better surgical outcomes and quality of life without showing any oncological inferiority compared to those treated with primary resection. The study aims to provide more data and useful information for the decision-making process in choosing the best treatment for obstructive left colon cancer.

## 1. Introduction

Colorectal cancer (CRC) is one of the most common cancers, and with 1.4 million new cases per year, it represents the second most prevalent cancer worldwide. The rising incidence in certain countries is indicative of lifestyle changes and their effects associated with “Westernization”, including factors such as obesity, lack of physical activity, alcohol consumption, high intake of red meat, and smoking [1]. Age remains the most significant unmodifiable risk factor for sporadic colon cancer, with nearly 70% of patients being over 65 years old; the disease is uncommon before the age of 40. However, data from Western registries indicate a growing incidence in the 40–44 age group [2]. The cumulative lifetime risk of developing CRC is approximately 6%, and it depends on factors that can be classified into lifestyle or behavioural characteristics and genetically determined factors [3,4,5,6,7].

CRC-related emergency clinical conditions present in a wide variety of patients, with an incidence from 7% to 40% according to the literature, although most studies cite an incidence of around 30% [8]. Among these emergencies, large bowel obstruction (LBO) accounts for nearly 80% (15–30% of CRC cases), while perforation represents the remaining 20% (1–10% of CRC cases) [9]. The sigmoid colon is the most common site for CRC-related obstruction, with 75% of tumours occurring distal to the splenic flexure [10].

CRC is not the only cause of LBO. Other mechanical causes include diverticular strictures, volvulus, and inflammatory bowel disease; however, in 50% of cases, LBO is due to a colorectal malignancy [11]. In 10–30% of these cases, LBO is the presenting condition [12,13,14], and when related to CRC, it is associated with a poorer long-term prognosis for patients with stage III cancers [15,16]. In absence of treatment, the most frequent complications include intestinal ischemia and perforation; for this reason, in LBO management, urgent decompression is required. Several decompression methods are available, and there is a great debate on the best management approach to choose for this acute condition due to the extremely heterogeneous results on the short- and long-term outcomes these methods may provide [8].

Surgical options include emergency resection of the primary tumour with an immediate colorectal anastomosis, potentially combined with a diverting loop ileostomy or, alternatively, a resection without an immediate recanalization (“Hartmann’s procedure”). Although the Hartmann’s procedure is widely used, it is associated with higher morbidity and mortality rates, and in up to 40% of cases, the stoma becomes permanent [14,17,18,19,20,21,22].

The endoscopic positioning of self-expanding metallic stents (SEMS) represents the main non-surgical method in LBO management. Since the mid-1990s, the use of these devices has proven to be effective in LBO resolution, and in selected patients, they can be used as a bridge-to-surgery method, thus allowing a more likely minimally invasive treatment of the tumour, reducing morbidity and the risk of a stoma [23].

Our study aims to describe and compare the surgical and oncological outcomes of the different treatment strategies for obstructing left colon cancer in our tertiary referral centre, also emphasizing the importance of quality of life after surgery, a parameter that patients often value more than others when deciding to undergo emergency surgery [24].

## 2. Materials and Methods

This is a monocentric, retrospective observational study of acutely (unplanned and non-elective presentation to the hospital for urgent or emergency reasons) presenting patients to our Emergency Departments treated for obstructive left colon cancer.

### 2.1. Study Population

#### 2.1.1. Inclusion Criteria

We included patients of both sexes, ≥18 years old, admitted to our emergency department in a period between 1 January 2019 and 15 September 2023, and who underwent abdominal CT scan that showed colonic obstruction due to obstructive left colonic cancer (sigmoid or descending colon) without distant metastases or perforation. All patients were treated with an endoscopic or surgical approach and presented with a colonic adenocarcinoma on postoperative histological examination.

The patients undergoing a surgical approach were indicated as the primary resection group (PR group), while the patients undergoing an endoscopic approach were indicated as the staged resection group (SR group).

#### 2.1.2. Exclusion Criteria

Locally advanced CRC (TNM T4) was considered as an exclusion criterion. We also excluded patients with abscess, perforation, or fistula associated with the main lesion. Pregnant and lactating women were excluded, too.

#### 2.1.3. Postoperative Follow-Up

After resection with curative intent, all patients entered the oncological monitoring program starting with chemotherapy when indicated according to institutional guidelines. We visited each patient 30 and 90 days after surgery and recorded postoperative complications according to Clavien–Dindo classification (CD) [25], number of readmissions, timing of chemotherapy initiation, and eventual stoma reversal. Restaging TC or a follow-up TC was carried out yearly and earlier if necessary. According to this, we performed a 1-year follow-up after surgery and reported the timing of chemotherapy initiation, eventual stoma reversal with relative stoma rate, oncological patient conditions (metastasis and recurrence), and survival. We also carried out a 3-year follow-up reporting metastasis rate, local recurrence rate, and overall survival. To each patient, the EQ-5D-5L test was administered to assess the quality of life after surgery; test items are reported in Figure 1 [26].

### 2.2. Colonic Stenting

All stenting procedures were performed by experienced endoscopists (>50 interventional endoscopic procedures). All stents used were uncovered and were positioned using the over-the-wire (OTW) technique under radiological guidance. The length and the diameter of the stent were chosen by the endoscopist according to the length of the tumour reported on the pre-procedural CT-scan. During the procedure, a biopsy of the tumour was performed. Clinical success was defined as resolution of acute obstruction with stool passage. No patients needed emergency surgery due to an adverse event from the stent. Primary tumour resection was performed from 7 to 51 days after placement of colonic stent.

### 2.3. Surgical Technique

All surgical procedures were performed in our high-volume tertiary centre (>170 colorectal surgery procedure per year) by experienced surgeons with a strong expertise in emergency surgery (minimum 50 emergency colonic resections) and minimally invasive colorectal surgery (minimum 60 procedure for a laparoscopic approach and minimum 40 procedures for a robotic approach). A laparoscopic approach was attempted for both emergency surgery and for surgery after stenting. A robotic approach was attempted whenever robotic equipment was available (DaVinci Xi^®^, Intuitive Surgical, Sunnyvale, CA, USA). Stoma formation was decided by the managing surgeon at the time of surgery.

### 2.4. Statistical Analysis

Univariate analysis was conducted to evaluate patients’ characteristics, intraoperative variables, and oncological variables. Categorical variables were reported as percentages and absolute numbers. The continuous variables with a non-normal distribution (according to the Shapiro–Wilk test) were reported using the median, while the continuous variables with a normal distribution were reported using mean and standard deviation. Categorical variables were analysed using Fisher’s exact test, non-normal distributed continuous variables using the Mann–Whitney test, and normally distributed continuous variables using the Student’s *t*-test. A *p*-value < 0.05 was considered statistically significant (two-sided). All statistical analyses were performed using SPSS VR Statistics v. 24.0 (IBM, Armonk, NY, USA).

### 2.5. Ethical Aspect and Publication Policy

This is a monocentric retrospective observational study; it meets and conforms to the standards outlined in the Declaration of Helsinki and Good Epidemiological Practices. All physicians involved in the patient recruitment and the producing process of this study are included in the research authorship.

## 3. Results

Seventy-two patients with non-metastatic, obstructing left colon cancer were included in the analysis, of whom 36 underwent PR, and 36 underwent SR. All the 36 patients that underwent SR had successful stenting procedures. Patients’ characteristics and clinicopathological data are summarised in Table 1. There were no significant differences in baseline characteristics between the PR and SR.

### 3.1. Surgical Outcomes

Postoperative surgical results are summarised in Table 2. In terms of immediate post-operative outcomes, there was no significant difference in the use of post-operative ICU (38.9% vs. 22.2%, *p* = 0.471).

About overall postoperative complications (Clavien–Dindo section), we reported two patients needing a Hartman’s procedure for anastomosis leak (CD IIIb) in the PR group (5.6%), while two patients were treated in ICU for acute respiratory distress (CD IVa) in the SR group (5.6%), all patients were then discharged. A statistical significance was reached for the post-operative mortality (CD V), which was higher in the PR group compared to the SR group (22.2% vs. 0%, *p* = 0.014). In this group, seven patients died from septic complications; another one died from an acute coronary syndrome.

The PR group had a significantly longer hospital stay compared to the SR group (median of 10 days vs. 7 days, *p* = 0.01).

A significant difference was reported in the surgical technique (*p* < 0.01); an open approach was preferred in the PR group (32% vs. 0%), while a laparoscopic approach was consistently more used in the SR group (30% vs. 2%). Two patients from the PR group (5.6%) and four patients from the SR group (11.1%) were approached with a laparoscopic technique then converted open, and two patients only in the SR group (5.6%) were approached with a robotic technique.

A significant difference was reported also for type of resection between the two groups (*p* < 0.01): Left hemicolectomy without diverting stoma was preferred in the SR (88.9%) compared to the PR group (16.7%); four patients in the SR group (11.1%) and eight patients in the PR group (22.2%) underwent a left hemicolectomy requiring a diverting stoma; Hartmann’s procedure was performed exclusively in the PR group (61.1%).

The post-operative stoma rate was significantly different between the two groups (*p* < 0.01), with a higher proportion of stomas in the PR group (83.3%) compared to the SR group (11.1%).

There were no significant differences in histopathological characteristics such as T stage (*p* = 0.177), N stage (*p* = 1), and TNM stage (0.258), between the two groups.

### 3.2. 90-Day Follow-Up Outcomes

The 90-day follow-up results are summarised in Table 3. All patients from the SR group were alive after a 90-day post-operative period, while twenty-eight of the thirty-six initial patients were followed in the PR group due to immediate post-operative mortality (CD V) in this group.

A Clavien–Dindo II complication, pneumonia for one patient and heart failure for another, occurred only in the SR group (5.6%, *p* = 1), and it required readmission with no significant differences between the two groups (*p* = 1); no other complications occurred in the two groups.

Sixteen patients in the SR group (44.4%) and two patients in the PR group (7.1%) started chemotherapy within 90 days from surgery, with a significant difference between the two groups (*p* = 0.042).

A statistical significance was maintained in the stoma rate, which was higher in the PR group (*p* < 0.01).

Two patients in both the SR group (50% of all stomas) and in the PR group (11.1% of all stomas) had stoma reversal with no significant difference between the two groups.

### 3.3. 1-Year Follow-Up Outcomes

The 1-year follow-up results are summarised in Table 4. No patients in the SR group started chemotherapy within 1 year, while six patients did in the PR group (21.4%), with no significant differences (*p* = 0.229). No patients in the SR group developed local recurrence, while four patients did in the PR group (14.3%), with no significant differences (*p* = 0.168). Two of these four patients in the PR group (8.3%) developed secondary cancer: One patient showed liver metastasis, and the other showed brain metastasis, and they both started chemotherapy within 1 year before secondary cancer was diagnosticated. Two patients developed liver metastasis in the SR group (5.6%, *p* = 1).

Regarding surgical outcomes, both patients with stoma in the SR group had a reversal surgery (100%), while only four patients in the PR group did (25%), with no significant differences between the two groups (*p* = 0.333). They started with a statistical significance in the number of stomas per group (*p* = 0.032).

Twelve patients in the PR group (42.9%) had a stoma after 1-year follow-up, while there were no patients with a stoma in the SR group (*p* < 0.01).

At 1 year after surgery, we found in the two groups the same patients that were alive at 90-day follow-up, determining a 1-year survival rate of 100%.

We analysed 1-year overall survival between the two groups with Kaplan–Meier curve and log-rank test, as reported in Figure 2.

### 3.4. 3-Year Follow-Up Outcomes

The 3-year follow-up results are summarized in Table 5. We could extend a 3-year follow up to eight patients in the SR group and to 16 patients in the PR group. No significant differences were found between the two groups for metastasis rate (*p* = 1) and local recurrence rate (*p* = 0.385). No significant differences were also found for overall survival after 3 years (*p* = 0.122).

We analysed 3-year overall survival between the two groups with Kaplan–Meier curve and log-rank test, as reported in Figure 3.

### 3.5. Quality of Life Outcomes

Quality of life was assessed with the EQ-5D-5L test. Table 6 summarises the quality of life outcomes. A significant difference was found between the SR group and the PR group for usual activities items (*p* = 0.048) and anxiety/depression items (*p* = 0.025). No significant differences were found for the other analysed items.

## 4. Discussion

For technical reasons involving surgeon and endoscopist expertise and due to the concerns about long-term oncological outcome, there is not a unique validated treatment for the left-sided malignant colorectal obstruction. Guideline recommendations may vary, and in recent years, they have evolved. In 2014 the European Society of Gastrointestinal Endoscopy (ESGE) made a strong recommendation against the use of SEMS as a bridge-to-surgery method in their guidelines [29]; however, in the 2020 updated guidelines, SEMS could be an option if discussed with patients presenting a potentially curable left-sided malignant colorectal obstruction [17]. According to the American Society of Colon and Rectal Surgeons (ASCRS), SEMS is recommended as a bridge-to-surgery method, together with urgent oncologic resection, based on moderate quality evidence [30]. The World Society of Emergency Surgery (WSES) guidelines consider SEMS safe only in selected cases in tertiary care centres and exclude its use as a part of the routine management of left-sided malignant obstruction [8].

The treatment of obstructive CRC is a topic discussed in the literature. The different options proposed in several studies have produced controversial results. The bridge-to-surgery approach (“staged resection”) is the chosen option for some authors, as it allows to carry the patients towards an elective setting, avoiding a full operation in an emergency situation [7]. Other authors prefer an immediate tumour resection (“primary resection”) to potentially achieve better oncological outcomes [3]. Nevertheless, the surgical options in case of primary resection include an immediate colorectal anastomosis (associated with a loop diverting stoma) or a resection without a prompt recanalization (Hartmann’s procedure). The latter operation is a very popular option, but the stoma becomes permanent in up to 40% of patients [14,18,19,20,21,22,29]. These considerations were reflected in our study as well, in which the SR group presented a lower stoma rate immediately after surgery (*p* < 0.01), at 90-day follow-up (*p* = 0.01), and at 1-year follow-up (*p* < 0.01) compared to the PR group, in which Hartmann’s procedure was the preferred technique. As reported in the literature [31], in our experience, the feasibility of subsequent resection surgery without protective stomas may also increase after SEMS positioning.

In terms of minimally invasive surgery, SEMS placement is associated with a higher feasibility of laparoscopic or robotic surgery, as reported by a recent metanalysis [32]; this was based on 53 studies comparing urgent colonic resection, surgical diversion as BTS, and/or SEMS as BTS; minimally invasive resections were performed, respectively, in 20% of cases for urgent colonic resection; 32.8% of cases for surgical diversion; and 48.2% after SEMS positioning. In our study, we reported in the SR group a significantly higher proportion (*p* < 0.01) of operations performed with a laparoscopic (83.3%) or robotic (5.6%) approach compared to the PR group (respectively, 5.6% for laparoscopic and 0% for robotic approach). This suggests greater confidence in using minimally invasive techniques in patients who had undergone SEMS positioning. Laparoscopic resection following SEMS is safe and feasible, and its use also reduces patients’ post-operative pain and provides a better recovery [33,34]. In our study, these post-operative benefits were reflected in a shorter median hospital stay for patients in the SR group compared to the PR group (7 vs. 10 days); this could be related to the possibility of converting an urgent operation into a controlled semi-elective resection through bridge-to-surgery techniques. Other factors involved in shortening the hospital stay could be found in the time gained for optimization of patients, hydration, and nutrition that bridge-to-surgery techniques may provide. The findings in this study concur with the existing literature on the short-term benefits of successful stenting, such as a higher rate of laparoscopic surgery, a lower rate of stoma creation, and a shorter hospital stay [18].

The complication rates reported in our study did not show differences in the two groups, except for the mortality rate that occurred exclusively in the PR group (22.2%). It must be specified that all patients dead after surgery underwent Hartmann’s procedure; they were all elderly people with an impairment of clinical condition due to both the acute obstruction consequences and the pre-existing diseases.

Despite all the benefits associated with a BTS strategy with SEMS, some concerns remain regarding the oncological safety of these devices. This is due to the high tumour dissemination and early metastasis associated with SEMS placement reported in several studies through a mechanism of “silent perforation” that can occur in up to 27% of resected specimens [35,36,37]. Peritoneal recurrence seems to be the most common site of distant metastasis, with a higher risk of recurrence in bridge-to-surgery patients [38,39]. Other authors have found SEMS safe since they did not report differences between them and other treatments in term of long-term oncological outcomes [40,41,42,43]. To better examine this aspect, the multicentre prospective study CROSCO-1, still in the development phase, aims to evaluate the surgical and oncological outcomes in patients treated with a primary resection compared with patients undergoing a bridge-to-surgery treatment [44].

In our study, between the groups we analysed, the SR group started chemotherapy within 90 days in a higher proportion of patients, but this did not express better oncological outcomes since metastasis occurred after 1 year from surgery in both groups, and local recurrence occurred only in the PR group but with no significant differences. A 3-year follow-up was conducted with no statistical differences in metastasis, local recurrence rate, and overall survival, but it was based on a smaller group of patients since we have been placing stents consistently only for the last 2 years in our centre. On this topic, in our study, we showed that patients who undergo a staged resection start chemotherapy before the others, but we did not find a direct influence on long-term oncological outcomes such as local recurrence and metastasis in the two groups.

An important outcome to evaluate was the patients’ quality of life after surgery. We determined it with the EQ-5D-5L test that showed a significantly better quality of life in the SR group: the “Usual activities” items and “Anxiety/Depression” items. These results may be due to a faster recovery after surgery due to the minimally invasive approach and due to a lower stoma rate, which exempts patients from the physical and psychological discomfort of stoma management. However, the literature is highly lacking on this aspect, and to our knowledge, there is only one study that analysed this parameter evaluating the mean global health status, as assessed with the QL2 subscale of the European Organization for Research and Treatment of Cancer (EORTC) quality-of-life questionnaires, in patients divided into an SEMS group and a primary resection group; unfortunately, the trial was concluded due to an increased high perforation rate (13%) in the SEMS group since centres with limited experience in SEMS placement were included in the study [45].

Currently, the wide range of therapeutic options for managing obstructive left-sided colorectal cancer (CRC) impacts the quality of scientific evidence regarding both short- and long-term surgical and oncological outcomes. The diversity in approaches is also influenced by the individual surgeon’s experience and the availability of operative endoscopy services. Also, the experience and expertise of each endoscopist may influence treatment decisions, and they may play a role in defining long-term outcomes [31,46].

## 5. Conclusions

SEMS followed by a staged resection is a feasible option in the treatment of obstructive CRC. It provides better surgical short-term outcomes with a better quality of life. It must be performed by trained endoscopists to reduce the possibility of a complication in SEMS positioning, such as bowel perforation. A minimally invasive approach, namely laparoscopic or robotic, is more likely to be used after SEMS positioning, and in most cases, a stoma can be avoided, bringing the patient back to normal life earlier.

Some concerns remain about long-term oncological outcomes since we could not extend the follow-up period to all patients, but further studies may give us more certain results.

However, we are aware that our work has some limitations. In fact, it is a retrospective, monocentric study with a not very large cohort size. Most patients were followed for a short period, and this might have affected the results from the analysis of the oncological and long-term survival data. Therefore, multicentre studies with a higher level of evidence and a longer follow-up, such as the CROSCO study [44], will be necessary to acquire more information, especially about long-term oncological outcomes.

## Figures and Tables

**Figure 1 cancers-16-03895-f001:**
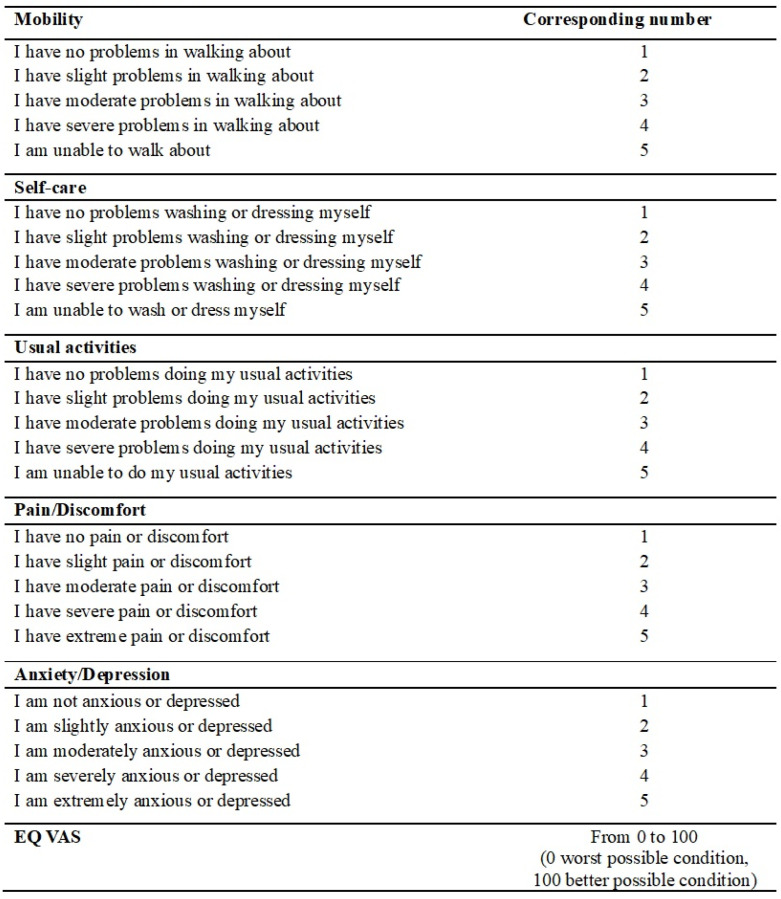
EQ-5D-5L test.

**Figure 2 cancers-16-03895-f002:**
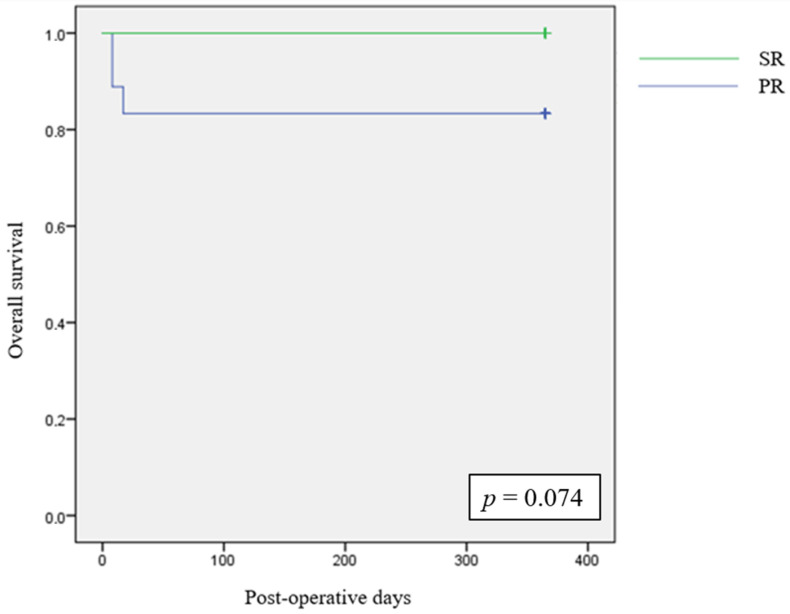
The 1-year Kaplan–Meier survival estimates.

**Figure 3 cancers-16-03895-f003:**
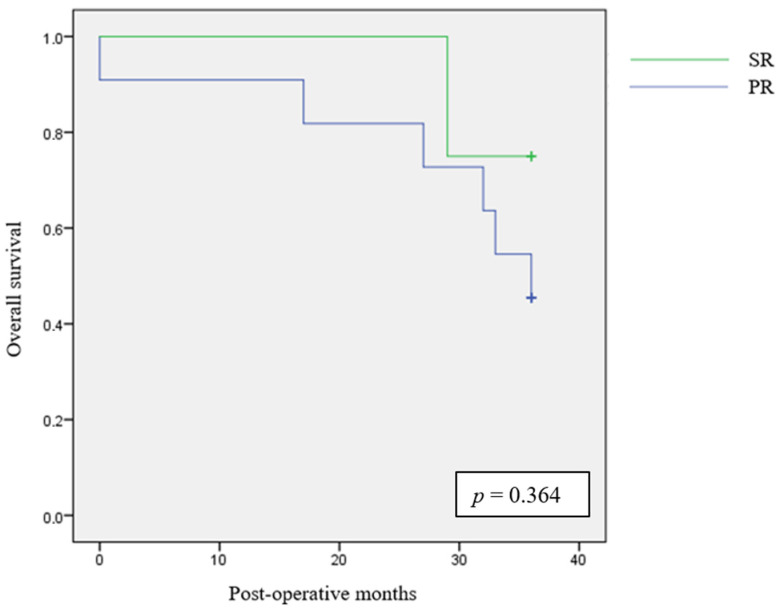
The 3-year Kaplan–Meier survival estimates.

**Table 1 cancers-16-03895-t001:** General Patient characteristics.

Variable	SR (n = 36)	PR (n = 36)	*p*-Value
Age, median (range)	79 (52–92)	81.5 (50–92)	0.437
Gender (%)			0.733
Female	16	12	
Male	20	24	
CCI (%)			0.688
1	0	0	
2	0	0	
3	2 (5.6)	2 (5.6)	
4	2 (5.6)	4 (11.1)	
5	12 (33.3)	4 (11.1)	
6	14 (38.9)	16 (44.4)	
7	4 (11.1)	4 (11.1)	
8	2 (5.6)	6 (16.7)	
BMI, mean (SD)	24.1 (3.7)	24.4 (4.6)	0.413
Diabetes (%)	6 (16.7)	10 (27.8)	0.691
Smoking (%)	6 (16.7)	4 (11.1)	1
Liver disease (%)	2 (5.6)	0	1
Kidney disease (%)	6 (16.7)	4 (11.1)	1
Past solid tumor (%)	8 (22,2)	6 (16.7)	1
Past abdominal surgery (%)	14 (38.9)	20 (55.6)	0.505
ASA (%)			0.181
I	0	0	
II	24 (66.7)	14 (38.9)	
III	12 (33.3)	20 (55.6)	
IV	0	2 (5.6)	
Location of tumor (%)			1
Sigmoid colon	24 (66.7)	24 (66.7)	
Descending colon	10 (27.8)	12 (33.3)	
Rectosigmoid	2 (5.6)	0	

CCI: Charlson Comorbidity Index [27]; BMI: Body Mass Index; ASA: American Society of Anaesthesiologist classification [28].

**Table 2 cancers-16-03895-t002:** Postoperative surgical results.

Variable	SR (n = 36)	PR (n = 36)	*p*-Value
Hospital stay, median (range)	7 (5–30)	10 (4–20)	**0.01**
Post-operative ICU (%)	14 (38.9)	8 (22.2)	0.471
Resection type (%)			**<0.01**
Hartmann’s procedure	0	22 (61.1)	
Left hemicolectomy with diverting stoma	4 (11.1)	8 (22.2)	
Left hemicolectomy without diverting stoma	32 (88.9)	6 (16.7)	
Surgical approach (%)			**<0.01**
Open	0	32 (88.9)	
Laparoscopic	30 (83.3)	2 (5.6)	
Laparoscopic converted open	4 (11.1)	2 (5.6)	
Robotic	2 (5.6)	0	
Robotic converted open	0	0	
Post-operative Clavien–Dindo (%)			
I	8 (22.2)	6 (16.7)	1
II	0	4 (11.1)	0.486
IIIa	0	0	-
IIIb	0	2 (5.6)	1
IVa	2 (5.6)	0	1
IVb	0	0	-
V (post-operative dead patients)	0	8 (22.2)	**0.014**
Stoma (%)	4 (11.1)	30 (83.3)	**<0.01**
T stage (%)			0.177
T2	2 (5.6)	10 (27.8)	
T3	34 (94.4)	23 (72.2)	
N stage (%)			1
N0	20 (55.6)	22 (61.1)	
N1	12 (33.3)	10 (27.8)	
N2	4 (11.1)	4 (11.1)	
TNM stage (%)			0.258
I	2 (5.6)	10 (27.8)	
II	18 (50)	12 (33.3)	
III	16 (44.4)	14 (38.9)	

Statistically significant values are given in bold; ICU: Intensive Care Unit.

**Table 3 cancers-16-03895-t003:** The 90-day follow-up results.

Variable	SR (n = 36)	PR (n = 28)	*p*-Value
90-day Clavien–Dindo (%)			
I	0	0	-
II	2 (5.6)	0	1
IIIa	0	0	-
IIIb	0	0	-
IVa	0	0	-
IVb	0	0	-
V	0	0	-
90-day readmission (%)	2 (5.6)	0	1
Started CHT within 90 days (%)	16 (44.4)	2 (7.1)	**0.042**
Stoma (%)	4 (11.1)	22 (78.6)	**<0.01**
Stoma reversed within 90 days (% of all stomas)	2 (50)	2 (9.1)	0.318

Statistically significant values are given in bold; CHT: Chemotherapy.

**Table 4 cancers-16-03895-t004:** The 1-year follow-up results.

Variable	SR (n = 36)	PR (n = 28)	*p*-Value
Started CHT within 1 year (%)	0	6 (21.4)	0.229
1-year metastasis (%)	2 (5.6)	2 (8.3)	1
1-year local recurrence (%)	0	4 (14.3)	0.168
Stoma (%)	2 (5.6)	20 (7.1)	**0.032**
Stoma reversed within 1 year (% of all stomas)	2 (100)	4 (20)	0.333
1-year stoma rate (%)	0	16 (57.1)	**<0.01**
1-year survival (%)	36 (100)	28 (100)	**-**

Statistically significant values are given in bold; CHT: Chemotherapy.

**Table 5 cancers-16-03895-t005:** The 3-year follow-up.

Variable	SEMS (n = 8)	PR (n = 16)	*p*-Value
3-year metastasis (%)	2 (25)	2 (12.5)	1
3-year local recurrence (%)	2 (25)	0	0.385
3-year survival (%)	6 (75)	10 (62.5)	0.122

**Table 6 cancers-16-03895-t006:** Overall quality of life evaluated by the EQ-5D-5L test.

Variable	SEMS (n = 32)	PR (n = 14)	*p*-Value
Mobility (%)			0.523
1	24 (75)	8 (57.1)	
2	2 (6.25)	4 (28.6)	
3	2 (6.25)	0	
4	2 (6.25)	0	
5	2 (6.25)	2 (14.3)	
Self-care (%)			0.573
1	26 (81.25)	12 (85.7)	
2	0	0	
3	4 (12.5)	0	
4	2 (6.25)	0	
5	0	2 (14.3)	
Usual activities (%)			**0.048**
1	26 (81.25)	12 (85.7)	
2	0	0	
3	4 (12.5)	0	
4	2 (6.25)	0	
5	0	2 (14.3)	
Pain/Discomfort			0.120
1	18 (56.25)	12 (85.7)	
2	10 (31.25)	0	
3	4 (12.5)	0	
4	0	0	
5	0	2 (14.3)	
Anxiety/Depression			**0.025**
1	12 (37.5)	8 (57.1)	
2	10 (31.25)	2 (14.3)	
3	6 (18.75)	0	
4	2 (6.25)	2 (14.3)	
5	2 (6.25)	2 (14.3)	
EQ-VAS, median (range)	85 (10–100)	70 (30–100)	0.893

Statistically significant values are given in bold.

## Data Availability

The data presented in this study are available on request from the corresponding author since they include sensitive information about the clinical history of the patients involved.

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
