# Peer review of "Upfront Surgery vs. Endoscopic Stenting Bridge to Minimally Invasive Surgery for Treatment of Obstructive Left Colon Cancer: Analysis of Surgical and Oncological Outcomes"

_cancers, 2024, doi:10.3390/cancers16233895_

Round 1

Reviewer 1 Report

Comments and Suggestions for Authors

The authors present a well written manuscript and a well conducted retrospective study. The topic is of relevance to our readership. The following queries must be addressed:

1. The authors must do a better job at emphasizing the relevance from their study and its contribution. If this is a novel type of study, they must state it so. If not, how does it contribute anything new compared to prior literature?

2. The experience from the surgery and GI team performing the procedures must be further described and detailed as well as the experience from the center.

3. How was the decision to operate with the robot vs the laparoscopic approach made. Do the authors think this posed any risk of bias? Can they explain the rationale?

4. What are the limitations of the study and what types of bias do the authors detect? How can they compensate for these weaknesses?

5. Can the authors offer any future directions and ideas for further research?

6. The conclusion must be summarized. It contains helpful information that belongs in the discussion.

Congratulations on a well conducted study.

Author Response

We thank you for the compliments, these make us happy and we gladly answer your questions. We tried to make our article clearer improving the text according to your suggestions. You will find below all the answers to your request. All modified or added sentences according to your request are highlighted in green, yellow and red highlighted sentences respond respectively to the editor's and the other reviewers' requests. Please see the attachment.

Comment 1: "The authors must do a better job at emphasizing the relevance from their study and its contribution. If this is a novel type of study, they must state it so. If not, how does it contribute anything new compared to prior literature?"

Response 1: We are aware that ours is not a new type of study. There are other works with similar characteristics in the literature, they are all cited. Unlike these, in our case, we also analyzed the patients' quality of life after surgery. We added a sentence to underline this aspect (page 3, introduction section, line 94-95). This concept was also remarked in the discussion section referring to another study (page 14, discussion section, lines 352-353). The added sentence and the cited section are highlighted in green.

Comment 2: "The experience from the surgery and GI team performing the procedures must be further described and detailed as well as the experience from the center."

Response 2: "We added the number of colo-rectal surgery procedures performed in our center (>170 per year) and the minimum number of procedures we must perform to be defined experienced (minimum 50 emergency colon resections for emergency surgery, minimum 60 procedures for laparoscopic approach, minimum 40 procedures for robotic approach, minimum 50 procedure for endoscopic approach) (page 4, colonic stenting section, lines 129-130; page 4 surgical technique section, from line 139 to line 143). Added sentences are highlighted in green.

Comment 3: "How was the decision to operate with the robot vs the laparoscopic approach made. Do the authors think this posed any risk of bias? Can they explain the rationale?"

Response 3: We did not really choose a laparoscopic approach or a robotic approach according to clinical or oncological criteria. We usually share robotic equipment (DaVinci Xi) with other specialists (Urologists, Gynecologists, Thoracic Surgeons and Pancreatic Surgeons) and we attempt a robotic approach whenever the equipment is available. We added a sentence to the text to state this concept (page 4, surgical technique section, lines 144-145). We do not think this could create a bias since, in our center, both laparoscopic and robotic approaches are extremely standardized providing similar results in terms of surgical and oncological safety. The added sentence is highlighted in green

Comment 4: "What are the limitations of the study and what types of bias do the authors detect? How can they compensate for these weaknesses?"

Response 4: "We listed all the weaknesses in the conclusion section referring to the monocentric retrospective nature of the study and the small sample of patients (page 14, conclusion section, lines 378-379). We found several biases, the first is about post-operative mortality which resulted higher in the PR group and this could be related to the initial clinical condition of these patients as stated in the text (page 13, discussion section, from line 324 to line 328); the other is about the 3-year follow-up that could be provided for only a part of the patients since we have been placing SEMS continuously for no more than two years and this may affect the long-term oncological results as stated in the text (page 14, discussion section, from line 343 to line 346). It is hard to compensate for these weaknesses and biases due to the retrospective nature of the study. All the cited sections are highlighted in green.

Comment 5: "Can the authors offer any future directions and ideas for further research?"

Response 5: We are aware that studies with higher levels of evidence must be provided to better understand the implications of stent placement. We believe that multicentric prospective studies like the CROSCO study, in which we are directly involved, could provide further information on all the aspects that our study could not analyze. (page 14, discussion section, from line 336 to 339; page 14, conclusion section, lines 381-382). Alle the cited sections are highlighted in green.

Comment 6: "The conclusion must be summarized. It contains helpful information that belongs in the discussion."

Response 6: We listed all the benefits about surgical outcomes provided by the bridge-to-surgery strategy (page 14, conclusion section, from line 370 to line 375). We summarized the non clear oncological benefits with a single statement (page 14, conclusion section, lines 376 -377 -378). All corrections are highlighted in green.

We hope you will find out article improved.

Kind regards

Reviewer 2 Report

Comments and Suggestions for Authors

Make corrections as attached

Author Response

Comment: "Make corrections as attached"

Response: Thank you for the suggested corrections. We included them in the article, you can find them highlighted in red. The editor asked us to modify some sections, all these changes are highlighted in yellow. The green highlighted sentences are modified according to other reviewers requests.

Please see the attachment. We hope you will find our article improved.

Kind regards

Round 2

Reviewer 1 Report

Comments and Suggestions for Authors

The researchers have done an excellent job at answering the queries with diligence and respect. The manuscript is appropriate for publication now in my view. Of course, the decision will be up to the editor in chief, but I approve. Congratulations to the writers.